# ROBUST TRAINING WITH ENSEMBLE CONSENSUS

**Jisoo Lee & Sae-Young Chung**
Korea Advanced Institute of Science and Technology
Daejeon, South Korea
`{jisoolee,schung}@kaist.ac.kr`

## ABSTRACT

Since deep neural networks are over-parameterized, they can memorize noisy examples. We address such a memorization issue in the presence of label noise. From the fact that deep neural networks cannot generalize to neighborhoods of memorized features, we hypothesize that noisy examples do not consistently incur small losses on the network under a certain perturbation. Based on this, we propose a novel training method called *Learning with Ensemble Consensus* (LEC) that prevents overfitting to noisy examples by removing them based on the consensus of an ensemble of perturbed networks. One of the proposed LECs, LTEC outperforms the current state-of-the-art methods on noisy MNIST, CIFAR-10, and CIFAR-100 in an efficient manner.

## 1 INTRODUCTION

Deep neural networks (DNNs) have shown excellent performance (Krizhevsky et al., 2012; He et al., 2016) on visual recognition datasets (Deng et al., 2009). However, it is difficult to obtain high-quality labeled datasets in practice (Wang et al., 2018a). Even worse, DNNs might not learn patterns from the training data in the presence of noisy examples (Zhang et al., 2016). Therefore, there is an increasing demand for robust training methods. In general, DNNs optimized with SGD first learn patterns relevant to clean examples under label noise (Arpit et al., 2017). Based on this, recent studies regard examples that incur small losses on the network that does not overfit noisy examples as clean (Han et al., 2018; Shen & Sanghavi, 2019). However, such small-loss examples could be noisy, especially under a high level of noise. Therefore, sampling trainable examples from a noisy dataset by relying on small-loss criteria might be impractical.

To address this, we find the method to identify noisy examples among small-loss ones based on well-known observations: (i) noisy examples are learned via memorization rather than via pattern learning and (ii) under a certain perturbation, network predictions for memorized features easily fluctuate, while those for generalized features do not. Based on these two observations, we hypothesize that out of small-loss examples, training losses of noisy examples would increase by injecting certain perturbation to network parameters, while those of clean examples would not. This suggests that examples that consistently incur small losses under multiple perturbations can be regarded as clean. This idea comes from an artifact of SGD optimization, thereby being applicable to any architecture optimized with SGD.

In this work, we introduce a method to perturb parameters to distinguish noisy examples from small-loss examples. We then propose a method to robustly train neural networks under label noise, which is termed *learning with ensemble consensus* (LEC). In LEC, the network is initially trained on the entire training set for a while and then trained on the intersection of small-loss examples of the ensemble of perturbed networks. We present three LECs with different perturbations and evaluate their effectiveness on three benchmark datasets with random label noise (Goldberger & Ben-Reuven, 2016; Ma et al., 2018), open-set noise (Wang et al., 2018b), and semantic noise. Our proposed LEC outperforms existing robust training methods by efficiently removing noisy examples from training batches.

## 2  RELATED WORK

**Generalization of DNNs.**   Although DNNs are over-parameterized, they have impressive generalization ability (Krizhevsky et al., 2012; He et al., 2016). Some studies argue that gradient-based optimization plays an important role in regularizing DNNs (Neyshabur et al., 2014; Zhang et al., 2016). Arpit et al. (2017) show that DNNs optimized with gradient-based methods learn patterns relevant to clean examples in the early stage of training. Since mislabeling reduces the correlation with other training examples, it is likely that noisy examples are learned via memorization. Therefore, we analyze the difference between generalized and memorized features to discriminate clean and noisy examples.

**Training DNNs with Noisy datasets.**   Label noise issues can be addressed by reducing negative impact of noisy examples. One direction is to train with a modified loss function based on the noise distribution. Most studies of this direction estimate the noise distribution prior to training as it is not accessible in general (Sukhbaatar et al., 2014; Goldberger & Ben-Reuven, 2016; Patrini et al., 2017; Hendrycks et al., 2018). Another direction is to train with modified labels using the current model prediction (Reed et al., 2014; Ma et al., 2018). Aside from these directions, recent work suggests a method of exploiting small-loss examples (Jiang et al., 2017; Han et al., 2018; Yu et al., 2019; Shen & Sanghavi, 2019) based on the generalization ability of DNNs. However, it is still hard to find clean examples by relying on training losses. This study presents a simple method to overcome such a problem of small-loss criteria.

## 3  ROBUST TRAINING WITH ENSEMBLE CONSENSUS

### 3.1  PROBLEM STATEMENT

Suppose that $\epsilon\%$ of examples in a dataset $\mathcal{D} := \mathcal{D}_{clean} \cup \mathcal{D}_{noisy}$ are noisy. Let $\mathcal{S}_{\epsilon,\mathcal{D},\theta}$ denote the set of $(100\text{-}\epsilon)\%$ small-loss examples of the network $f$ parameterized by $\theta$ out of examples in $\mathcal{D}$. Since it is generally hard to learn only all clean examples especially on the highly corrupted training set, it is problematic to regard all examples in $\mathcal{S}_{\epsilon,\mathcal{D},\theta}$ as being clean. To mitigate this, we suggest a simple idea: to find noisy examples among examples in $\mathcal{S}_{\epsilon,\mathcal{D},\theta}$.

### 3.2  LEARNING WITH ENSEMBLE CONSENSUS (LEC)

Since noisy examples are little correlated with other training examples, they are likely to be learned via memorization. However, DNNs cannot generalize to neighborhoods of the memorized features. This means that even if training losses of noisy examples are small, they can be easily increased under a certain perturbation $\delta$, *i.e.*, for $(x, y) \in \mathcal{D}_{noisy}$,

$$(x, y) \in \mathcal{S}_{\epsilon,\mathcal{D},\theta} \Rightarrow (x, y) \notin \mathcal{S}_{\epsilon,\mathcal{D},\theta+\delta}.$$

Unlike noisy examples, the network $f$ trained on the entire set $\mathcal{D}$ can learn patterns from some clean examples in the early stage of training. Thus, their training losses are consistently small in the presence of the perturbation $\delta$, *i.e.*, for $(x, y) \in \mathcal{D}_{clean}$,

$$(x, y) \in \mathcal{S}_{\epsilon,\mathcal{D},\theta} \Rightarrow (x, y) \in \mathcal{S}_{\epsilon,\mathcal{D},\theta+\delta}.$$

This suggests that noisy examples can be identified from the inconsistency of losses under certain perturbation $\delta$. Based on this, we regard examples in the intersection of $(100\text{-}\epsilon)\%$ small-loss examples of an ensemble of $M$ networks generated by adding perturbations $\delta_1, \delta_2, ..., \delta_M$ to $\theta$, *i.e.*,

$$\cap_{m=1}^{M} \mathcal{S}_{\epsilon,\mathcal{D},\theta+\delta_m}$$

as clean. We call it **ensemble consensus filtering** because examples are selected via ensemble consensus. With this filtering, we develop a training method termed **learning with ensemble consensus** (LEC) described in Algorithms 1 and 2. Both algorithms consist of *warming-up* and *filtering* processes. The difference between these two lies in the filtering process. During the filtering process of Algorithm 1, the network is trained on the intersection of $(100\text{-}\epsilon)\%$ small-loss examples of $M$ networks within a mini batch $\mathcal{B}$. Therefore, the number of examples updated at once is changing.

We can encourage more stable training with a fixed number of examples to be updated at once as described in Algorithm 2. During the filtering process of Algorithm 2, we first obtain the intersection of small-loss examples of $M$ networks within a full batch $\mathcal{D}$ at each epoch. We then sample a subset of batchsize from the intersection and train them at each update like a normal SGD.

---

**Algorithm 1** LEC

**Require:** noisy dataset $\mathcal{D}$ with noise ratio $\epsilon\%$, duration of warming-up $T_w$, # of networks used for filtering $M$, perturbation $\delta$
1: Initialize $\theta$ randomly
2: **for** epoch $t = 1 : T_w$ **do** ▶ Warming-up process
3:     **for** mini-batch index $b = 1 : \frac{|\mathcal{D}|}{\text{batchsize}}$ **do**
4:        Sample a subset of batchsize $\mathcal{B}_b$ from a full batch $\mathcal{D}$
5:        $\theta \leftarrow \theta - \alpha \nabla_\theta \frac{1}{|\mathcal{B}_b|} \sum_{(x,y) \in \mathcal{B}_b} CE(f_\theta(x), y)$
6:     **end for**
7: **end for**
8: **for** epoch $t = T_w + 1 : T_{end}$ **do** ▶ Filtering process
9:     **for** mini-batch index $b = 1 : \frac{|\mathcal{D}|}{\text{batchsize}}$ **do**
10:       Sample a subset of batchsize $\mathcal{B}_b$ from a full batch $\mathcal{D}$
11:       **for** $m = 1 : M$ **do**
12:          $\theta_m = \theta + \delta_{m,b,t}$ ▷ Adding perturbation
13:          $\mathcal{S}_{\epsilon, \mathcal{B}_b, \theta_m} := (100 - \epsilon)\%$ small-loss examples of $f_{\theta_m}$ within a mini batch $\mathcal{B}_b$
14:       **end for**
15:       $\mathcal{B}_b' = \cap_{m=1}^{M} \mathcal{S}_{\epsilon, \mathcal{B}_b, \theta_m}$ ▷ Ensemble consensus filtering
16:       $\theta \leftarrow \theta - \alpha \nabla_\theta \frac{1}{|\mathcal{B}_b'|} \sum_{(x,y) \in \mathcal{B}_b'} CE(f_\theta(x), y)$
17:     **end for**
18: **end for**

---

**Algorithm 2** LEC-full

**Require:** noisy dataset $\mathcal{D}$ with noise ratio $\epsilon\%$, duration of warming-up $T_w$, # of networks used for filtering $M$, perturbation $\delta$
1: Initialize $\theta$ randomly
2: **for** epoch $t = 1 : T_w$ **do** ▶ Warming-up process
3:     **for** mini-batch index $b = 1 : \frac{|\mathcal{D}|}{\text{batchsize}}$ **do**
4:        Sample a subset of batchsize $\mathcal{B}_b$ from a full batch $\mathcal{D}$
5:        $\theta \leftarrow \theta - \alpha \nabla_\theta \frac{1}{|\mathcal{B}_b|} \sum_{(x,y) \in \mathcal{B}_b} CE(f_\theta(x), y)$
6:     **end for**
7: **end for**
8: **for** epoch $t = T_w + 1 : T_{end}$ **do** ▶ Filtering process
9:     **for** $m = 1 : M$ **do**
10:       $\theta_m = \theta + \delta_{m,t}$ ▷ Adding perturbation
11:       $\mathcal{S}_{\epsilon, \mathcal{D}, \theta_m} := (100 - \epsilon)\%$ small-loss examples of $f_{\theta_m}$ within a full batch $\mathcal{D}$
12:     **end for**
13:     $\mathcal{D}_t' = \cap_{m=1}^{M} \mathcal{S}_{\epsilon, \mathcal{D}, \theta_m}$ ▷ Ensemble consensus filtering
14:     **for** mini-batch index $b = 1 : \frac{|\mathcal{D}_t'|}{\text{batchsize}}$ **do**
15:       Sample a subset of batchsize $\mathcal{B}_b'$ from $\mathcal{D}_t'$
16:       $\theta \leftarrow \theta - \alpha \nabla_\theta \frac{1}{|\mathcal{B}_b'|} \sum_{(x,y) \in \mathcal{B}_b'} CE(f_\theta(x), y)$
17:     **end for**
18: **end for**

---

## 3.3 Perturbation to identify noisy examples

Now we aim to find a perturbation $\delta$ to be injected to discriminate memorized features from generalized ones. We present three LECs with different perturbations in the following. The pseudocodes can be found in Section A.1.3.

- **Network-Ensemble Consensus (LNEC)**: Inspired by the observation that an ensemble of networks with the same architecture is correlated during generalization and is decorrelated during memorization (Morcos et al., 2018), the perturbation $\delta$ comes from the difference between $M$ **networks**. During the warming-up process, $M$ networks are trained independently. During the filtering process, $M$ networks are trained on the intersection of $(100-\epsilon)\%$ small-loss examples of $M$ networks.

- **Self-Ensemble Consensus (LSEC)**: We focus on the relationship between Morcos et al. (2018) and Lakshminarayanan et al. (2017): network predictions for memorized features are uncertain and those for generalized features are certain. Since the uncertainty of predictions also can be captured by multiple stochastic predictions (Gal & Ghahramani, 2016), the perturbation $\delta$ comes from the difference between $M$ **stochastic predictions of a single network**.[1] During the filtering process, the network is trained on the intersection of $(100-\epsilon)\%$ small-loss examples obtained with $M$ stochastic predictions.

- **Temporal-Ensemble Consensus (LTEC)**: Inspired by the observation that during training, atypical features are more easily forgetful compared to typical features (Toneva et al., 2018), the perturbation $\delta$ comes from the difference between **networks at current and preceding epochs**. During the filtering process, the network is trained on the intersection of $(100-\epsilon)\%$ small-loss examples at the current epoch $t$ and preceding $\min(M - 1, t - 1)$ epochs. We collect $(100-\epsilon)\%$ small-loss examples at the preceding epochs, rather than network parameters to reduce memory usage.

---

[1]As in Gal & Ghahramani (2016), the stochasticity of predictions is caused by stochastic operations such as dropout (Srivastava et al., 2014).

# 4 EXPERIMENTS

In this section, we show (i) the effectiveness of three perturbations at removing noisy examples from small-loss examples and (ii) the comparison of LEC and other existing methods under various annotation noises.

## 4.1 EXPERIMENTAL SETUP

**Annotation noise.** We study random label noise (Goldberger & Ben-Reuven, 2016; Ma et al., 2018), open-set noise (Wang et al., 2018b), and semantic noise. To generate these noises, we use MNIST (LeCun et al., 1998), CIFAR-10/100 (Krizhevsky et al., 2009) that are commonly used to assess the robustness. For each benchmark dataset, we only corrupt its training set, while leaving its test set intact for testing. The details can be found in Section A.1.1.

- **Random label noise.** Annotation issues can happen in easy images as well as hard images (Wang et al., 2018a). This is simulated in two ways: **sym-$\epsilon$%** and **asym-$\epsilon$%**. For sym-$\epsilon$%, $\epsilon$% of the entire set are randomly mislabeled to one of the other labels and for asym-$\epsilon$%, each label $i$ of $\epsilon$% of the entire set is changed to $i + 1$. We study four types: sym-20% and asym-20% to simulate a low level of noise, and sym-60% and asym-40% to simulate a high level of noise.

- **Open-set noise.** In reality, annotated datasets may contain out-of-distribution (OOD) examples. As in Yu et al. (2019), to make OOD examples, images of $\epsilon$% examples randomly sampled from the original dataset are replaced with images from another dataset, while labels are left intact. SVHN (Netzer et al., 2011) is used to make open-set noise of CIFAR-100, and ImageNet-32 (Chrabaszcz et al., 2017) and CIFAR-100 are used to make open-set noise of CIFAR-10. We study two types: 20% and 40% open-set noise.

- **Semantic noise.** In general, images with easy patterns are correctly labeled, while images with ambiguous patterns are obscurely mislabeled. To simulate this, we select the top $\epsilon$% most uncertain images and then flip their labels to the confusing ones. The uncertainty of each image is computed by the amount of disagreement between predictions of networks trained with clean dataset as in Lakshminarayanan et al. (2017).[2] Then, the label of each image is assigned to the label with the highest value of averaged softmax outputs of the networks trained with a clean dataset except for its ground-truth label. We study two types: 20% and 40% semantic noise.

**Architecture and optimization.** Unless otherwise specified, we use a variant of 9-convolutional layer architecture (Laine & Aila, 2016; Han et al., 2018). All parameters are trained for 200 epochs with Adam (Kingma & Ba, 2014) with a batch size of 128. The details can be found in Section A.1.2.

**Hyperparameter.** The proposed LEC involves three hyperparameters: duration of warming-up $T_w$, noise ratio $\epsilon$%, and the number of networks used for filtering $M$. Unless otherwise specified, $T_w$ is set to 10, and $M$ is set to 5 for random label noise and open-set noise, and 10 for semantic noise. We assume that a noise ratio of $\epsilon$% is given. Further study can be found in Section 5.2.

**Evaluation.** We use two metrics: test accuracy and label precision (Han et al., 2018). At the end of each epoch, test accuracy is measured as the ratio of correctly predicted test examples to all test examples, and label precision is measured as the ratio of clean examples used for training to examples used for training. Thus, for both metrics, higher is better. For methods with multiple networks, the averaged values are reported. We report peak as well as final accuracy because a small validation set may be available in reality.

For each noise type, every method is run four times with four random seeds, *e.g.*, four runs of Standard on CIFAR-10 with sym-20%. A noisy dataset is randomly generated and initial network parameters are randomized for each run of both random label noise and open-set noise. Note that four noisy datasets generated in four runs are the same for all methods. On the other hand, semantic noise is generated in a deterministic way. Thus, only initial network parameters are randomized for each run of semantic noise.

---

[2]The uncertainty of image $x$ is defined by $\sum_{n=1}^{N} KL(f(x; \theta_n) || \frac{1}{N} \sum_{n=1}^{N} f(x; \theta_n))$ where $f(; \theta)$ denotes softmax output of network parameterized by $\theta$. Here, $N$ is set to 5 as in Lakshminarayanan et al. (2017).

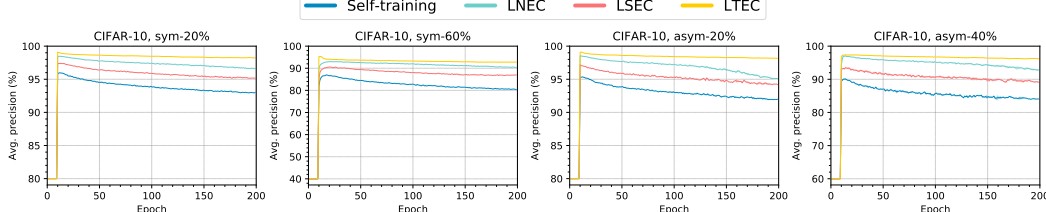

Figure 1: **Label precision** (%) of Self-training and three LECs on CIFAR-10 with **random label noise**. We plot the average as a solid line and the standard deviation as a shadow around the line.

Table 1: Average of **final/peak test accuracy** (%) of Self-training and three LECs on CIFAR-10 with **random label noise**. The best is highlighted in **bold**.

| Dataset | Noise type | Self-training | LNEC | LSEC | LTEC |
|---|---|---|---|---|---|
| CIFAR-10 | sym-20% | 84.96/85.02 | 86.72/86.78 | 85.42/85.63 | 88.18/**88.28** |
| | sym-60% | 73.99/74.35 | 79.61/79.64 | 76.73/76.92 | 80.38/**80.52** |
| | asym-20% | 85.02/85.24 | 86.90/87.11 | 85.44/85.64 | 88.86/**88.93** |
| | asym-40% | 78.84/79.66 | 84.01/84.48 | 80.74/81.49 | 86.36/**86.50** |

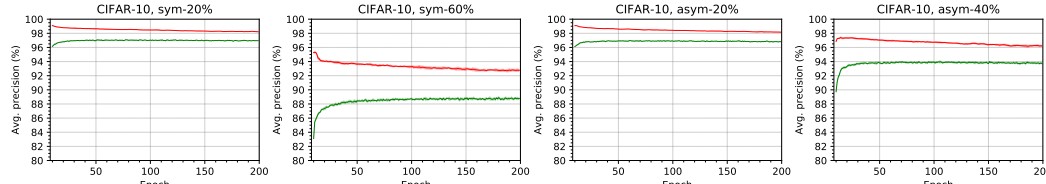

Figure 2: **Label precision** (%) of **small-loss examples of the current network** (in green) and **the intersection of small-loss examples of the current and preceding networks** (in red) during running LTEC on CIFAR-10 with random label noise. We report the precision from epoch 11 when the filtering process starts.

## 4.2 EFFECTIVENESS OF LECs AT IDENTIFYING NOISY EXAMPLES

**Comparison with Self-training.**  In Section 3.1, we argue that $(100\text{-}\epsilon)\%$ small-loss examples may be corrupted. To show this, we run LEC with $M = 1$, which is a method of training on $(100\text{-}\epsilon)\%$ small-loss examples. Note that this method is similar to the idea of Jiang et al. (2017); Shen & Sanghavi (2019). We call it ***Self-training*** for simplicity. Figure 1 shows the label precision of Self-training is low especially under the high level of noise, *i.e.*, sym-60%. Compared to Self-training, three LECs are trained on higher precision data, achieving higher test accuracy as shown in Table 1. Out of these three, LTEC performs the best in both label precision and test accuracy.

**Noisy examples are removed through ensemble consensus filtering.**  In LTEC, at every batch update, we first obtain $(100\text{-}\epsilon)\%$ small-loss examples of the current network and then train on the intersection of small-loss examples of the current and preceding networks. We plot label precisions of small-loss examples of the current network (in green) and the intersection (in red) during running LTEC on CIFAR-10 with random noise in Figure 2. We observe that label precision of the intersection is always higher, indicating that noisy examples are removed through ensemble consensus filtering.

## 4.3 COMPARISON WITH STATE-OF-THE-ART METHODS

**Competing methods.**  The competing methods include a regular training method: ***Standard***, a method of training with corrected labels: ***D2L*** (Ma et al., 2018), a method of training with modified loss function based on the noise distribution: ***Forward*** (Patrini et al., 2017), and a method of exploiting small-loss examples: ***Co-teaching*** (Han et al., 2018). We tune all the methods individually as described in Section A.1.4.

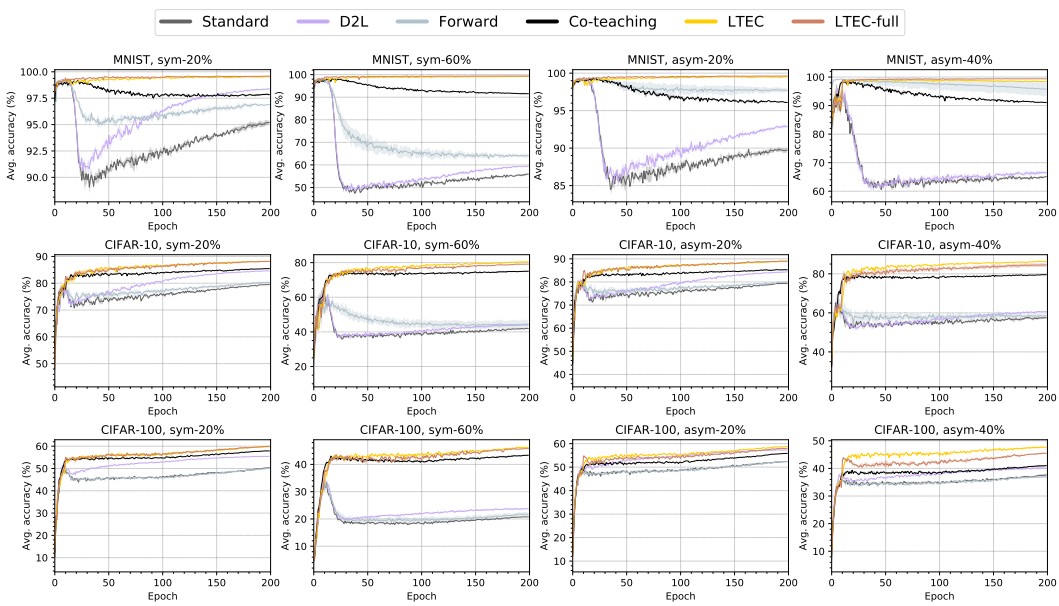

Figure 3: **Test accuracy** (%) of different algorithms on MNIST/CIFAR with **random label noise**.

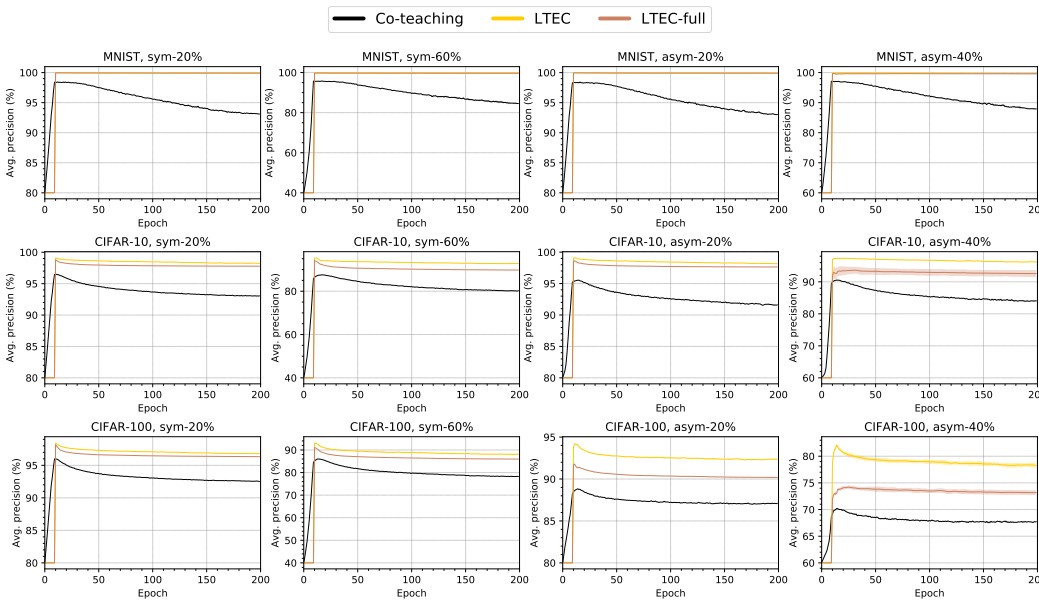

Figure 4: **Label precision** (%) of different algorithms on MNIST/CIFAR with **random label noise**.

**Results on MNIST/CIFAR with random label noise.** The overall results can be found in Figures 3 and 4, and Table 2. We plot the average as a solid line and the standard deviation as a shadow around the line. Figure 3 states that the test accuracy of D2L increases at the low level of label noise as training progresses, but it does not increase at the high level of label noise. This is because D2L puts large weights on given labels in the early stage of training even under the high level of noise. Forward shows its strength only in limited scenarios such as MNIST. Co-teaching does not work well on CIFAR-100 with asym-40%, indicating that its cross-training scheme is vulnerable to small-loss examples of a low label precision (see Figure 4). Unlike Co-teaching, our methods attempt to remove noisy examples in small-loss examples. Thus, on CIFAR-100 with asym-40% noise, both LTEC and LTEC-full surpass Co-teaching by a wide margin of about 6% and 5%, respectively.

Table 2: Average of **final/peak test accuracy** (%) of different algorithms on MNIST/CIFAR with **random label noise**. The best is highlighted in **bold**.

| Dataset | Noise type | Standard | D2L | Forward | Co-teaching | LTEC | LTEC-full |
|---|---|---|---|---|---|---|---|
| MNIST | sym-20% | 95.21/99.36 | 98.38/99.35 | 96.88/99.29 | 97.84/99.24 | 99.52/99.58 | 99.58/**99.64** |
| | sym-60% | 55.88/98.50 | 59.40/98.37 | 64.03/98.26 | 91.52/98.53 | 99.16/99.25 | 99.38/**99.44** |
| | asym-20% | 89.74/99.32 | 92.88/99.41 | 97.71/99.52 | 96.11/99.40 | 99.49/99.59 | 99.61/**99.66** |
| | asym-40% | 65.13/96.58 | 66.44/96.99 | 95.76/**99.51** | 91.10/98.81 | 98.47/99.32 | 99.40/99.48 |
| CIFAR-10 | sym-20% | 79.50/80.74 | 84.60/84.68 | 80.29/80.91 | 85.46/85.52 | 88.18/88.28 | 88.16/**88.31** |
| | sym-60% | 41.91/65.06 | 44.10/65.26 | 44.38/61.89 | 75.01/75.19 | 80.38/**80.52** | 79.13/79.26 |
| | asym-20% | 79.24/81.39 | 84.27/84.40 | 79.89/82.08 | 85.24/85.44 | 88.86/88.93 | 89.04/**89.14** |
| | asym-40% | 57.50/68.77 | 60.63/67.46 | 58.53/67.19 | 79.53/80.19 | 86.36/**86.50** | 84.56/84.69 |
| CIFAR-100 | sym-20% | 50.28/50.89 | 55.47/55.58 | 50.01/50.58 | 57.87/57.94 | 59.73/59.82 | 59.91/**59.98** |
| | sym-60% | 20.79/34.26 | 23.72/34.89 | 21.78/34.01 | 43.36/43.68 | 46.24/**46.43** | 45.77/45.89 |
| | asym-20% | 52.40/52.42 | 57.31/57.53 | 52.44/52.56 | 55.88/55.91 | 58.72/**58.86** | 58.05/58.16 |
| | asym-40% | 37.64/37.66 | 40.12/40.37 | 36.95/37.61 | 40.99/41.01 | 47.70/**47.82** | 45.49/45.55 |

Table 3: Average of **final/peak test accuracy** (%) of different algorithms on CIFAR with **open-set noise**. The best is highlighted in **bold**.

| Dataset + Open-set | Noise type | Standard | D2L | Forward | Co-teaching | LTEC | LTEC-full |
|---|---|---|---|---|---|---|---|
| CIFAR-10 + CIFAR-100 | 20% | 86.74/86.83 | 89.42/**89.49** | 86.87/86.96 | 88.58/88.61 | 88.69/88.82 | 89.07/89.11 |
| | 40% | 82.64/82.71 | 85.32/85.41 | 82.57/82.68 | 86.18/86.22 | 86.37/**86.41** | 86.26/86.33 |
| CIFAR-10 + ImageNet-32 | 20% | 88.27/88.36 | 90.60/**90.64** | 88.24/88.29 | 88.99/89.06 | 89.15/89.24 | 89.34/89.42 |
| | 40% | 85.90/85.99 | 87.91/**87.95** | 85.84/85.99 | 86.99/87.03 | 86.63/86.78 | 87.00/87.12 |
| CIFAR-100 + SVHN | 20% | 59.08/59.19 | 62.89/**62.98** | 58.99/59.08 | 60.69/60.75 | 61.65/61.78 | 61.87/61.98 |
| | 40% | 53.32/53.35 | 56.30/56.38 | 53.18/53.30 | 56.45/56.52 | 56.95/57.18 | 57.77/**57.90** |

Table 4: Average of **final/peak test accuracy** (%) of different algorithms on CIFAR with **semantic noise**. The best is highlighted in **bold**.

| Dataset | Noise type | Standard | D2L | Forward | Co-teaching | LTEC | LTEC-full |
|---|---|---|---|---|---|---|---|
| CIFAR-10 | 20% | 81.29/81.36 | 83.96/83.99 | 81.10/81.23 | 83.53/83.56 | 84.48/**84.66** | 84.48/84.58 |
| | 40% | 71.64/74.36 | 74.72/74.94 | 71.38/73.47 | 76.61/76.89 | 75.52/76.52 | 76.57/**78.06** |
| CIFAR-100 | 20% | 56.88/56.96 | 60.23/**60.40** | 56.60/56.74 | 58.45/58.50 | 58.75/58.78 | 58.73/58.80 |
| | 40% | 49.56/49.69 | 53.04/53.19 | 49.57/49.69 | 52.96/52.98 | 52.58/52.78 | 53.15/**54.18** |

**Results on CIFAR with open-set noise.** The overall results can be found in Table 3. All the methods including LTEC and LTEC-full perform well under open-set noise. We speculate that this is due to a low correlation between open-set noisy examples. This is supported by the results on CIFAR-10, *i.e.*, all the methods perform better on ImageNet-32 noise than on CIFAR-100 noise, as ImageNet-32 has more classes than CIFAR-100. Similar to poorly annotated examples, it is hard for deep networks to learn patterns relevant to out-of-distribution examples during the warming-up process. Therefore, those examples can be removed from training batches through ensemble consensus filtering.

**Results on CIFAR with semantic noise.** The overall results can be found in Table 4. The semantically generated noisy examples are highly correlated with each other, making it difficult to filter out those examples through ensemble consensus. We use 10 as the value of $M$ for semantic noise because ensemble consensus with a bigger $M$ is more conservative. On CIFAR with semantic noise, LTEC and LTEC-full perform comparably or best, compared to the other methods. Of the two, LTEC-full performs better on 40% semantic noise due to its training stability.

## 5 DISCUSSION

### 5.1 HARD-TO-CLASSIFY BUT CLEAN EXAMPLES

It is hard to learn all clean examples during the warming-up process. Therefore, clean examples with large losses may be excluded from training batches during the filtering process. However, we expect that the number of clean examples used for training would increase gradually as training

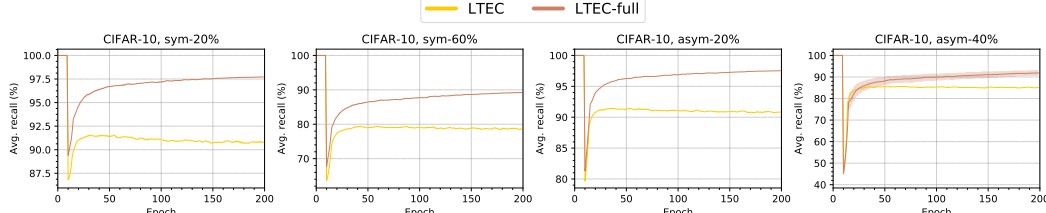

Figure 5: **Recall** (%) of LTEC and LTEC-full on CIFAR-10 with **random label noise**. We plot the average as a solid line and the standard deviation as a shadow around the line.

Table 5: Average of **final/peak test accuracy** (%) of LTEC with varying the number of networks used for filtering $M$. The best is highlighted in **bold**.

| Dataset | Noise type | LTEC | | | | | |
|---|---|---|---|---|---|---|---|
| | | $M = 1$ | $M = 3$ | $M = 5$ | $M = 7$ | $M = 9$ | $M = \infty$ |
| CIFAR-10 | sym-20% | 84.96/85.02 | 87.68/87.78 | 88.18/88.28 | 88.63/88.77 | 88.79/**88.87** | 86.57/86.62 |
| | sym-60% | 73.99/74.35 | 79.73/79.80 | 80.38/**80.52** | 80.39/80.45 | 80.28/80.39 | 71.63/71.86 |
| | asym-20% | 85.02/85.24 | 87.85/88.15 | 88.86/88.93 | 88.96/89.07 | 88.99/**89.11** | 85.55/85.59 |
| | asym-40% | 78.84/79.66 | 85.44/85.59 | 86.36/86.50 | 86.78/**86.82** | 86.59/86.63 | 77.30/77.40 |

Table 6: Average of **final/peak test accuracy** (%) of Co-teaching and LTEC with estimates of noise ratio (simulated). The best is highlighted in **bold**.

| Dataset | Noise type | under-estimated $(0.9\epsilon)$ | | correctly estimated $(\epsilon)$ | | over-estimated $(1.1\epsilon)$ | |
|---|---|---|---|---|---|---|---|
| | | Co-teaching | LTEC | Co-teaching | LTEC | Co-teaching | LTEC |
| CIFAR-10 | sym-20% | 84.51/84.58 | 87.93/88.08 | 85.46/85.52 | 88.18/88.28 | 86.40/86.45 | 88.72/**88.75** |
| | sym-60% | 70.47/73.11 | 77.98/78.22 | 75.01/75.19 | 80.38/**80.52** | 79.15/79.17 | 79.34/79.45 |
| | asym-20% | 84.61/84.73 | 88.15/88.39 | 85.24/85.44 | 88.86/88.93 | 86.41/86.57 | 89.04/**89.22** |
| | asym-40% | 76.14/77.41 | 84.42/84.52 | 79.53/80.19 | 86.36/86.50 | 82.19/82.63 | 86.93/**86.96** |

proceeds since LEC allows the network to learn from patterns clean examples without overfitting. To confirm this, we measure recall defined by the ratio of clean examples used for training to all clean examples at the end of each epoch during running LTEC and LTEC-full. As expected, recalls of both LTEC and LTEC-full sharply increase in the first 50 epochs as described in Figure 5. Pre-training (Hendrycks et al., 2019) prior to the filtering process may help to prevent the removal of clean examples from training batches.

## 5.2 ABLATION STUDY

**The number of networks used for filtering.** During the filtering process of LEC, we use only the intersection of small-loss examples of $M$ perturbed networks for training. This means that the number of examples used for training highly depends on $M$. To understand the effect of $M$, we run LTEC with varying $M$ on CIFAR-10 with random label noise. In particular, the range of $M$ is {1, 3, 5, 7, 9, $\infty$}. Table 5 shows that a larger $M$ is not always lead to better performance. This is because too many examples may be removed from training batches as $M$ increases. Indeed, the total number of examples used for training is critical for the robustness as claimed in Rolnick et al. (2017); Li et al. (2017).

**Noise ratio.** In reality, only a poorly estimated noise ratio may be accessible. To study the effect of poor noise estimates, we run LTEC on CIFAR-10 with random label noise using a bit lower and higher values than the actual noise ratio as in Han et al. (2018). We also run Co-teaching that requires the noise ratio for comparison. The overall results can be found in Table 6. Since it is generally difficult to learn all clean examples, training on small-loss examples selected using the over-estimated ratio (*i.e.*, $1.1\epsilon$) is often helpful in both Co-teaching and LTEC. In contrast, small-loss examples selected using the under-estimated ratio may be highly corrupted. In this case, LTEC is robust to the estimation error of noise ratio, while Co-teaching is not. Such robustness of LTEC against noise estimation error comes from ensemble consensus filtering.

Table 7: Average of **final/peak** test accuracy (%) of Standard and LTEC with ResNet. The best is highlighted in **bold**.

| Dataset | Noise Type | Standard (ResNet) | LTEC (ResNet) |
|---------|-----------|-------------------|---------------|
| CIFAR-10 | sym-20% | 81.31/85.30 | 89.01/**89.12** |
|  | sym-60% | 61.94/72.80 | 81.46/**81.66** |
|  | asym-20% | 81.93/87.32 | 88.90/**89.04** |
|  | asym-40% | 62.76/77.10 | 86.62/**86.85** |

**Applicability to different architecture.** The key idea of LEC is rooted in the difference between generalizaton and memorization, *i.e.*, the ways of learning clean examples and noisy examples in the early SGD optimization (Arpit et al., 2017). Therefore, we expect that LEC would be applicable to any architecture optimized with SGD. To support this, we run Standard and LTEC with ResNet-20 (He et al., 2016). The architecture is optimized based on Chollet et al. (2015), achieving the final test accuracy of 90.67% on clean CIFAR-10. Here, $T_w$ is set to 30 for the optimization details. Table 7 shows LTEC (ResNet) beats Standard (ResNet) in both peak and final accuracies, as expected.

## 6 CONCLUSION

This work presents the method of generating and using the ensemble for robust training. We explore three simple perturbation methods to generate the ensemble and then develop the way of identifying noisy examples through ensemble consensus on small-loss examples. Along with growing attention to the use of small-loss examples for robust training, we expect that our ensemble method will be useful for such training methods.

### ACKNOWLEDGMENTS

We thank Changho Suh, Jinwoo Shin, Su-Young Lee, Minguk Jang, and anonymous reviewers for their great suggestions. This work was supported by the ICT R&D program of MSIP/IITP. [2016-0-00563, Research on Adaptive Machine Learning Technology Development for Intelligent Autonomous Digital Companion]

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

## A  APPENDIX

### A.1  IMPLEMENTATION DETAILS

#### A.1.1  ANNOTATION NOISES

- **Random label noise**: For sym-$\epsilon$%, $\epsilon$% of the entire set are randomly mislabeled to one of the other labels and for asym-$\epsilon$%, each label $i$ of $\epsilon$% of the entire set is changed to $i + 1$. The corruption matrices of sym-$\epsilon$% and asym-$\epsilon$% are described in Figures A1a and A1b, respectively.
- **Open-set noise**: For $\epsilon$% open-set noise, images of $\epsilon$% examples randomly sampled from the original dataset are replaced with images from external sources, while labels are left intact. For CIFAR-10 with open-set noise, we sample images from 75 classes of CIFAR-100 (Abbasi et al., 2018) and 748 classes of ImageNet (Oliver et al., 2018) to avoid sampling similar images with CIFAR-10.
- **Semantic noise**: For semantic noise, we choose uncertain images and then mislabel them ambiguously. In Figure A2, we see that clean examples have simple and easy images, while noisy examples have not. Also, its corruption matrix (see Figure A1c) describes the similarity between classes, *e.g.*, cat and dog, car and truck, etc.

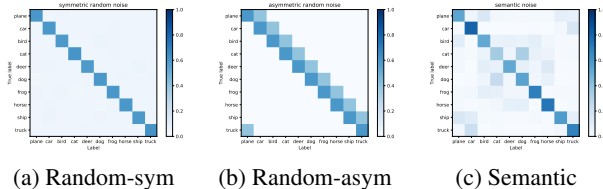

| (a) Random-sym | (b) Random-asym | (c) Semantic |

Figure A1: Corruption matrices of CIFAR-10 with random label noise and semantic noise.

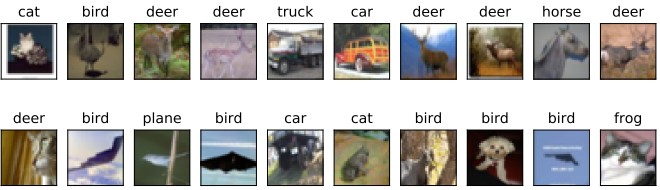

Figure A2: **Clean examples** (top) and **noisy examples** (bottom) randomly sampled from CIFAR-10 with 20% **semantic** noise. We observe that noisy examples contain atypical features and are semantically mislabeled.

### A.1.2 ARCHITECTURE AND OPTIMIZATION DETAILS

The 9-convolutional layer architecture used in this study can be found in Table A1. The network is optimized with Adam (Kingma & Ba, 2014) with a batchsize of 128 for 200 epochs. The initial learning rate $\alpha$ is set to 0.1. The learning rate is linearly annealed to zero during the last 120 epochs for MNIST and CIFAR-10, and during the last 100 epochs for CIFAR-100. The momentum parameters $\beta_1$ and $\beta_2$ are set to 0.9 and 0.999, respectively. $\beta_1$ is linearly annealed to 0.1 during the last 120 epochs for MNIST and CIFAR-10, and during the last 100 epochs for CIFAR-100. The images of CIFAR are divided by 255 and are whitened with ZCA. Additional regularizations such as data augmentation are not applied. The results on clean MNIST, CIFAR-10, and CIFAR-100 can be found in Table A2.

Table A1: 9-conv layer architecture.

| Input image |
| --- |
| Gaussian noise ($\sigma = 0.15$) |
| $3 \times 3$ conv, 128, padding = 'same' 
 batch norm, LReLU ($\alpha = 0.01$) 
 $3 \times 3$ conv, 128, padding = 'same' 
 batch norm, LReLU ($\alpha = 0.01$) 
 $3 \times 3$ conv, 128, padding = 'same' 
 batch norm, LReLU ($\alpha = 0.01$) 
 $2 \times 2$ maxpooling, padding = 'same' 
 dropout (drop rate = 0.25) |
| $3 \times 3$ conv, 256, padding = 'same' 
 batch norm, LReLU ($\alpha = 0.01$) 
 $3 \times 3$ conv, 256, padding = 'same' 
 batch norm, LReLU ($\alpha = 0.01$) 
 $3 \times 3$ conv, 256, padding = 'same' 
 batch norm, LReLU ($\alpha = 0.01$) 
 $2 \times 2$ maxpooling, padding = 'same' 
 dropout (drop rate = 0.25) |
| $3 \times 3$ conv, 512, padding = 'valid' 
 batch norm, LReLU ($\alpha = 0.01$) 
 $3 \times 3$ conv, 256, padding = 'valid' 
 batch norm, LReLU ($\alpha = 0.01$) 
 $3 \times 3$ conv, 128, padding = 'valid' 
 batch norm, LReLU ($\alpha = 0.01$) |
| global average pooling 
 fc (128 → # of classes) |

Table A2: Avg ($\pm$ stddev) of **final test accuracy** of a regular training on clean MNIST, CIFAR-10, and CIFAR-100.

| Dataset | MNIST | CIFAR-10 | CIFAR-100 |
| --- | --- | --- | --- |
| Test accuracy | 99.60±0.02 | 90.59±0.15 | 64.38±0.20 |

### A.1.3 PSEUDOCODES FOR LECs

We present three LECs with different perturbations in Section 3.3. The pseudocodes for LNEC, LSEC, LTEC, and LTEC-full are described in the following. In LTEC-full, we obtain small-loss examples utilized for filtering from the second epoch to encourage its stability.

---

**Algorithm A1** LNEC

---

**Require:** noisy dataset $\mathcal{D}$ with noise ratio $\epsilon\%$, duration of warming-up $T_w$, The number of networks used for filtering $M$
1:   Initialize $\theta_1, \theta_2, ..., \theta_M$ randomly
2:   **for** epoch $t = 1 : T_w$ **do**          ▶ Warming-up process
3:      **for** mini-batch index $b = 1 : \frac{|\mathcal{D}|}{\text{batchsize}}$ **do**
4:         Sample a subset of batchsize $\mathcal{B}_b$ from a full batch $\mathcal{D}$
5:         **for** network index $m = 1 : M$ **do**
6:            $\theta_m \leftarrow \theta_m - \alpha \nabla_{\theta_m} \frac{1}{|\mathcal{B}_b|} \sum_{(x,y) \in \mathcal{B}_b} CE(f_{\theta_m}(x), y)$
7:         **end for**
8:      **end for**
9:   **end for**
10:   **for** epoch $t = T_w + 1 : T_{end}$ **do**          ▶ Filtering process
11:      **for** mini-batch index $b = 1 : \frac{|\mathcal{D}|}{\text{batchsize}}$ **do**
12:         Sample a subset of batchsize $\mathcal{B}_b$ from a full batch $\mathcal{D}$
13:         **for** network index $m = 1 : M$ **do**
14:            $\mathcal{S}_{\epsilon, \mathcal{B}_b, \theta_m} := (100 - \epsilon)\%$ small-loss examples of $f_{\theta_m}$ within $\mathcal{B}_b$
15:         **end for**
16:         $\mathcal{B}_b' = \cap_{m=1}^M \mathcal{S}_{\epsilon, \mathcal{B}_b, \theta_m}$         ▷ Network-ensemble consensus filtering
17:         **for** network index $m = 1 : M$ **do**
18:            $\theta_m \leftarrow \theta_m - \alpha \nabla_{\theta_m} \frac{1}{|\mathcal{B}_b'|} \sum_{(x,y) \in \mathcal{B}_b'} CE(f_{\theta_m}(x), y)$
19:         **end for**
20:      **end for**
21:   **end for**

---

**Algorithm A2** LSEC

---

**Require:** noisy dataset $\mathcal{D}$ with noise ratio $\epsilon\%$, duration of warming-up $T_w$, The number of networks used for filtering $M$
1:   Initialize $\theta$ randomly
2:   **for** epoch $t = 1 : T_w$ **do**          ▶ Warming-up process
3:      **for** mini-batch index $b = 1 : \frac{|\mathcal{D}|}{\text{batchsize}}$ **do**
4:         Sample a subset of batchsize $\mathcal{B}_b$ from a full batch $\mathcal{D}$
5:         $\theta \leftarrow \theta - \alpha \nabla_\theta \frac{1}{|\mathcal{B}_b|} \sum_{(x,y) \in \mathcal{B}_b} CE(f_\theta(x), y)$
6:      **end for**
7:   **end for**
8:   **for** epoch $t = T_w + 1 : T_{end}$ **do**          ▶ Filtering process
9:      **for** mini-batch index $b = 1 : \frac{|\mathcal{D}|}{\text{batchsize}}$ **do**
10:         Sample a subset of batchsize $\mathcal{B}_b$ from a full batch $\mathcal{D}$
11:         **for** forward pass index $m = 1 : M$ **do**
12:            $\theta_m = \theta + \delta_m$ where $\delta_m$ comes from the stochasticity of network architecture
13:            $\mathcal{S}_{\epsilon, \mathcal{B}_b, \theta_m} := (100 - \epsilon)\%$ small-loss examples of $f_{\theta_m}$ within $\mathcal{B}_b$
14:         **end for**
15:         $\mathcal{B}_b' = \cap_{m=1}^M \mathcal{S}_{\epsilon, \mathcal{B}_b, \theta_m}$         ▷ Self-ensemble consensus filtering
16:         $\theta \leftarrow \theta - \alpha \nabla_\theta \frac{1}{|\mathcal{B}_b'|} \sum_{(x,y) \in \mathcal{B}_b'} CE(f_\theta(x), y)$
17:      **end for**
18:   **end for**

---

---

**Algorithm A3** LTEC

---

**Require:** noisy dataset $\mathcal{D}$ with noise ratio $\epsilon\%$, duration of warming-up $T_w$, The number of networks used for filtering $M$
1: Initialize $\theta$ randomly
2: **for** epoch $t = 1 : T_{end}$ **do**
3:      $\mathcal{P}_t = \varnothing$
4:      **for** mini-batch index $b = 1 : \frac{|\mathcal{D}|}{\text{batchsize}}$ **do**
5:          Sample a subset of batchsize $\mathcal{B}_b$ from a full batch $\mathcal{D}$
6:          $\mathcal{S}_{\epsilon,\mathcal{B}_b,\theta} := (100 - \epsilon)\%$ small-loss examples of $f_\theta$ within $\mathcal{B}_b$
7:          $\mathcal{P}_t \leftarrow \mathcal{P}_t \cup \mathcal{S}_{\epsilon,\mathcal{B}_b,\theta}$
8:          **if** $t < T_w + 1$ **then**                                   ▶ Warming-up process
9:             $\theta \leftarrow \theta - \alpha \nabla_\theta \frac{1}{|\mathcal{B}_b|} \sum_{(x,y) \in \mathcal{B}_b} CE(f_\theta(x), y)$
10:         **else**                                           ▶ Filtering process
11:             **if** $t = 1$ **then**
12:                 $\mathcal{B}_b' = \mathcal{S}_{\epsilon,\mathcal{B}_b,\theta}$
13:             **else if** $t < M$ **then**
14:                 $\mathcal{B}_b' = \mathcal{P}_1 \cap \mathcal{P}_2 \cap ... \cap \mathcal{P}_{t-1} \cap \mathcal{S}_{\epsilon,\mathcal{B}_b,\theta}$
15:             **else**
16:                 $\mathcal{B}_b' = \mathcal{P}_{t-(M-1)} \cap \mathcal{P}_{t-(M-2)} \cap ... \cap \mathcal{P}_{t-1} \cap \mathcal{S}_{\epsilon,\mathcal{B}_b,\theta}$      ▷ Temporal-ensemble consensus filtering
17:             **end if**
18:             $\theta \leftarrow \theta - \alpha \nabla_\theta \frac{1}{|\mathcal{B}_b'|} \sum_{(x,y) \in \mathcal{B}_b'} CE(f_\theta(x), y)$
19:         **end if**
20:      **end for**
21: **end for**

---

**Algorithm A4** LTEC-full

---

**Require:** noisy dataset $\mathcal{D}$ with noise ratio $\epsilon\%$, duration of warming-up $T_w$, The number of networks used for filtering $M$
1: Initialize $\theta$ randomly
2: **for** mini-batch index $b = 1 : \frac{|\mathcal{D}|}{\text{batchsize}}$ **do**
3:      Sample a subset of batchsize $\mathcal{B}_b$ from a full batch $\mathcal{D}$
4:      $\theta \leftarrow \theta - \alpha \nabla_\theta \frac{1}{|\mathcal{B}_b|} \sum_{(x,y) \in \mathcal{B}_b} CE(f_\theta(x), y)$
5: **end for**
6: **for** epoch $t = 2 : T_{end}$ **do**
7:      $\mathcal{P}_t := (100 - \epsilon)\%$ small-loss examples of $f_\theta$ within $\mathcal{D}$      ▷ Small-loss examples are computed from the 2nd epoch
8:      **if** $t < T_w + 1$ **then**                                     ▶ Warming-up process
9:          **for** mini-batch index $b = 1 : \frac{|\mathcal{D}|}{\text{batchsize}}$ **do**
10:             Sample a subset of batchsize $\mathcal{B}_b$ from a full batch $\mathcal{D}$
11:             $\theta \leftarrow \theta - \alpha \nabla_\theta \frac{1}{|\mathcal{B}_b|} \sum_{(x,y) \in \mathcal{B}_b} CE(f_\theta(x), y)$
12:          **end for**
13:      **else**                                           ▶ Filtering process
14:          **if** $t < M + 1$ **then**
15:             $\mathcal{D}_t' = \mathcal{P}_2 \cap \mathcal{P}_3 \cap ... \cap \mathcal{P}_{t-1} \cap \mathcal{P}_t$
16:          **else**
17:             $\mathcal{D}_t' = \mathcal{P}_{t-(M-1)} \cap \mathcal{P}_{t-(M-2)} \cap ... \cap \mathcal{P}_{t-1} \cap \mathcal{P}_t$      ▷ Temporal-ensemble consensus filtering
18:          **end if**
19:          **for** mini-batch index $b = 1 : \frac{|\mathcal{D}_t'|}{\text{batchsize}}$ **do**
20:             Sample a subset of batchsize $\mathcal{B}_b'$ from $\mathcal{D}_t'$
21:             $\theta \leftarrow \theta - \alpha \nabla_\theta \frac{1}{|\mathcal{B}_b'|} \sum_{(x,y) \in \mathcal{B}_b'} CE(f_\theta(x), y)$
22:          **end for**
23:      **end if**
24: **end for**

---

### A.1.4 COMPETING METHODS

The competing methods include a regular training method: ***Standard***, a method of training with corrected labels: ***D2L*** (Ma et al., 2018), a method of training with modified loss function based on the noise distribution: ***Forward*** (Patrini et al., 2017), and a method of exploiting small-loss examples: ***Co-teaching*** (Han et al., 2018). We tune all the methods individually as follows:

- **Standard** : The network is trained using the cross-entropy loss.
- **D2L**: The input vector of a fully connected layer in the architecture is used to measure the LID estimates. The parameter involved with identifying the turning point, window size $W$ is set to 12. The network is trained using original labels until the turning point is found and then trained using the bootstrapping target with adaptively tunable mixing coefficient.
- **Forward**: Prior to training, the corruption matrix $C$ where $C_{ji} = \mathbb{P}(y = i | y_{true} = j)$ is estimated based on the $97th$ percentile of probabilities for each class on MNIST and CIFAR-

10, and the $100th$ percentile of probabilities for each class on CIFAR-100 as in Hendrycks et al. (2018). The network is then trained using the corrected labels for 200 epochs.

- **Co-teaching**: Two networks are employed. At every update, they select their small-loss examples within a minibatch and then provide them to each other. The ratio of selected examples based on training losses is linearly annealed from 100% to (100-$\epsilon$)% over the first 10 epochs.

## A.2 COMPLEXITY ANALYSIS

We compute space complexity as the number of network parameters and computational complexity as the number of forward and backward passes. Here we assume that early stopping is not used and the noise ratio of $\epsilon$% is given. Note that the computational complexity of each method depends on its hyperparameter values, *e.g.*, the duration of the warming-up process $T_w$ and the noise ratio $\epsilon$%. The analysis is reported in Table A3. Our proposed LTEC is the most efficient because it can be implemented with a single network based on Section A.1.3 and only a subset of the entire training set is updated after the warming-up process.

Table A3: **Complexity analysis**: $M$ indicates the number of networks for filtering in LECs.

| Complexity | Standard | Self-training | Co-teaching | LNEC | LSEC | LTEC/LTEC-full |
|---|---|---|---|---|---|---|
| **Space complexity** | | | | | | |
| # of network parameters | $m$ | $m$ | $2m$ | $Mm$ | $m$ | $m$ |
| **Computational complexity** | | | | | | |
| # of forward passes | $n$ | $n$ | $2n$ | $Mn$ | $Mn$ | $n$ |
| # of backward passes | $n$ | $\leq n$ | $\leq 2n$ | $\leq Mn$ | $\leq n$ | $\leq n$ |

## A.3 ADDITIONAL RESULTS

### A.3.1 RESULTS OF LTEC WITH $M = \infty$

Figure A3 shows that ensemble consensus filtering with too large $M$ removes clean examples from training batches in the early stage of the filtering process. Unlike LTEC with $M = 5$, the recall of LTEC with $M = \infty$ does not increase as training proceeds, suggesting that its generalization performance is not enhanced. This shows that a larger $M$ does not always lead to better performance. We expect that pre-training (Hendrycks et al., 2019) prior to the filtering process helps to reduce the number of clean examples removed by ensemble consensus filtering regardless of $M$.

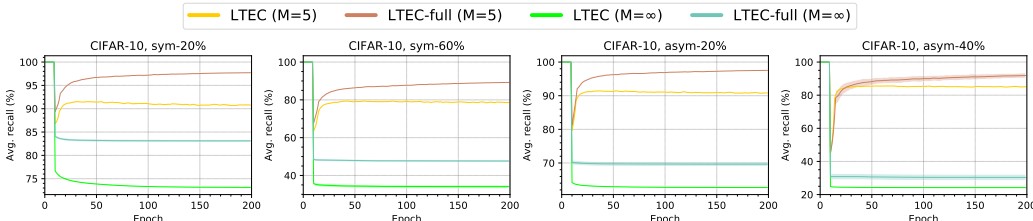

Figure A3: **Recall** (%) of LTECs with varying $M$ on CIFAR-10 with **random label noise**.

