# OpenReview forum: "Robust training with ensemble consensus"
_ICLR.cc/2020/Conference — Accept (Poster)_

### Official Review · AnonReviewer1 · 2019-10-14
**Official Blind Review #1**

**Rating:** 3

**Review:**

Caveat: I admit that I am not incredibly familiar with this particular research area, so I am probably not the best person to review this paper.  I am not certain what part of the matching and bidding process ended up with this paper assigned to me.

This paper presents a method for filtering out noisy training examples, assuming that there is a known percentage of label noise present in the data.  The gist of the method is to repeatedly perturb the weights in the network slightly, and only accept as training data examples that consistently get small loss across all perturbations.  The intuition is that the loss for noisily labeled examples will not be stable under these perturbations, so using an ensemble of perturbed models should find the label noise.  This intuition seems reasonable enough, and the method seems straightforward.

The reason I am giving a "weak reject" score is largely because the experiments seem weak to me.  It seems pretty unrealistic to randomly corrupt 20% or more of your data.  In what scenario will you actually have data that has 20%+ label noise?  If you actually have one, why not use that to show the effectiveness of your method, instead of an artificial setting?  You have shown that your method achieves better performance in a contrived setting, but in order for it to actually be _useful_, it needs to work on real data, which hasn't been shown.

One example of a real scenario where you might have a very high degree of label noise is in weakly-supervised semantic parsing.  This paper (https://openreview.net/forum?id=ryxjnREFwH), for example, uses confidence thresholding (though without an ensemble) to filter the training data.  It might be fruitful for you to try demonstrating the benefits of your method on this kind of task.

Minor issues:

Section 3.2 presents math suggesting that the perturbations are additive, with deltas drawn from some distribution.  Section 3.3 then presents the actual perturbation methods, which seem difficult to characterize as drawing a delta from a distribution.

Section 4.3 - citations for the baseline methods should be in the main paper, not only in the appendix.

Some things weren't explained well enough for the paper to be self-contained.  For example, the description of LSEC says that there are "stochastic operations", but these are not described anywhere, even with a simple sentence.  Instead the reader must refer to another paper.

**Experience Assessment:**

I do not know much about this area.

**Review Assessment: Checking Correctness Of Derivations And Theory:**

I did not assess the derivations or theory.

**Review Assessment: Checking Correctness Of Experiments:**

I assessed the sensibility of the experiments.

**Review Assessment: Thoroughness In Paper Reading:**

I read the paper at least twice and used my best judgement in assessing the paper.

---

> ### Author Response · Authors · 2019-11-15
> **Response to review#1**
>
> Thank you for your detailed feedback and suggestions. We would like to answer your questions as follows:
>
> Q1. The reason I am giving a "weak reject" score is largely because the experiments seem weak to me.  It seems pretty unrealistic to randomly corrupt 20% or more of your data.  In what scenario will you actually have data that has 20%+ label noise?
>
> >> It is common to use labeling tools to obtain labeled datasets without human intervention. However, in this case, label noise is likely to occur. For example, MS-celeb-1M(v1) contains approximately 50% (estimated) label noise according to [1].
>
> Q2. If you actually have one, why not use that to show the effectiveness of your method, instead of an artificial setting?  You have shown that your method achieves better performance in a contrived setting, but in order for it to actually be _useful_, it needs to work on real data, which hasn't been shown.
>
> >> We used the synthetic noisy datasets because it is easy to analyze the results. Due to this property, many robust training algorithms [2-5] have verified their effectiveness on random noise and open-set noise studied in our paper. We agree that evaluating our algorithm on actual datasets will be of great help in demonstrating its effectiveness. However, we did not have much time to conduct experiments with actual noisy datasets. Instead, we added the results on *e% semantic noise* generated as follows:
>
> “We first select the top e% most uncertain images from CIFAR and then mislabel those selected images ambiguously.”
>
> We believe that this semantic noise can mimic real noisy datasets because images with easy patterns are correctly labeled, while images with ambiguous patterns are obscurely mislabeled. We observed this through visualization of clean and noisy examples randomly sampled from the noisy dataset in Figure A2. Also, its corruption matrix describes the similarity of classes as seen in Figure A1. The details of semantic noise can be found in Section 4.1. On CIFAR with semantic noise, our proposed LEC performed comparably or best, compared to the other methods.
>
> Q3. (minor issue) Section 3.2 presents math suggesting that the perturbations are additive, with deltas drawn from some distribution.  Section 3.3 then presents the actual perturbation methods, which seem difficult to characterize as drawing a delta from a distribution.
>
> >> Sorry for the confusion. We defined some notions in Section 3.1 and clarified Section 3.2 and 3.3 using the notions in our revised paper.
>
> Q4. (minor issue) Section 4.3 - citations for the baseline methods should be in the main paper, not only in the appendix.
>
> >> We cited the relevant papers in the main text as well as in the appendix.
>
> Q5. (minor issue) Some things weren't explained well enough for the paper to be self-contained.
>
> >> Thanks for pointing it out.  We tried to make the revised paper self-contained. In particular, we added the explanation of “stochastic operation” to the paper as follows:
> the stochasticity of predictions is caused by stochastic operations “such as dropout”.
>
> * Reference
> [1] Wang et al., The devil of face recognition is in the Noise, ECCV 2018.
> [2] Han et al., Co-teaching: Robust Training of Deep Neural Networks with Extremely Noisy Labels, NeurIPS 2018.
> [3] Yu et al., How does disagreement help generalization against label corruption?, ICML 2019.
> [4] Lee et al., Robust Inference via Generative Classifiers for Handling Noisy Labels, ICML 2019.
> [5] Shu et al., Meta-Weight-Net: Learning an Explicit Mapping For Sample Weighting, NeurIPS 2019.
>
> Thanks again for taking the time to review.

---

### Official Review · AnonReviewer2 · 2019-10-22
**Official Blind Review #2**

**Rating:** 6

**Review:**

In this paper, the authors proposed to identify noisy training examples using ensemble consensus. The authors argued and demonstrated through numeric studies that, to the contrary of some earlier work, training examples with low training loss are not necessarily mislabeled. Rather, the authors hypothesized that examples with high noise require memorization, which is sensitive to perturbations. Thus, the authors proposed to identify and subsequently remove those examples from training by looking at the loss after small perturbations to the model parameters. Examples with consistently low training loss are retained for training. The authors also provided several alternatives of perturbations, including examining the consensus between an ensemble of networks, between multiple stochastic predictions, or between predictions from prior training epochs. Finally, the authors demonstrated the performance of their procedures using numerical studies.

This paper is well motivated and clearly presented. The idea of identifying noisy examples through ensemble consensus is novel and plausible. The numerical studies are relatively comprehensive and in-depth. I think this is a solid conference paper.

Some questions:
1. The authors proposed to perturb model parameters in order to find noisy training examples. Is there any reason that the authors did not perturb feature values in order to find noisy training examples? I would suppose that memorized examples are sensitive to feature value perturbation.
2. The intersection of small-loss examples in Line 13 of Algorithm 1 could be much smaller than (1 - epsilon / 100) * Bb, and could vary in size throughout training. Are there computationally efficient methods that can guarantee the size of the intersection is stable and not too small? I suppose that we do not want the mini-batch size to vary too much throughout training.
3. How to distinguish hard-to-classify, high-loss examples from high-noise, low-loss examples in Line 11 of Algorithm 1? Using the proposed algorithm, in addition to the noisy low-loss examples, we will also remove those hard-to-classify examples, which is arguably undesirable.
3. The authors should clearly discuss the computational complexity and space complexity of their proposals.

**Experience Assessment:**

I have read many papers in this area.

**Review Assessment: Checking Correctness Of Derivations And Theory:**

I carefully checked the derivations and theory.

**Review Assessment: Checking Correctness Of Experiments:**

I assessed the sensibility of the experiments.

**Review Assessment: Thoroughness In Paper Reading:**

I read the paper at least twice and used my best judgement in assessing the paper.

---

> ### Author Response · Authors · 2019-11-15
> **Response to review#2**
>
> Thank you for your detailed feedback on our paper. We would like to answer your questions as follows:
>
> Q1. The authors proposed to perturb model parameters in order to find noisy training examples. Is there any reason that the authors did not perturb feature values in order to find noisy training examples? I would suppose that memorized examples are sensitive to feature value perturbation.
>
> >> We thought that it would be more complicated to find noisy examples by perturbing features. But this is an interesting idea and we hope this to be developed in future work.
>
> Q2. The intersection of small-loss examples in Line 13 of Algorithm 1 could be much smaller than (1 - epsilon / 100) * Bb, and could vary in size throughout training. Are there computationally efficient methods that can guarantee the size of the intersection is stable and not too small? I suppose that we do not want the mini-batch size to vary too much throughout training.
>
> >> We can encourage more stable training with fixed batchsize as follows:
> At each epoch, we obtain the intersection of small-loss examples of networks within *a full batch*. At each update, we sample a subset of batchsize from the intersection and train them like a normal SGD.
> This method (termed *LEC-full*) was added to our revised paper. Also, we reported the performance of *LEC-full* evaluated on various noisy datasets.
>
> Q3. How to distinguish hard-to-classify, high-loss examples from high-noise, low-loss examples in Line 11 of Algorithm 1? Using the proposed algorithm, in addition to the noisy low-loss examples, we will also remove those hard-to-classify examples, which is arguably undesirable.
>
> >> As you mentioned, hard-to-classify examples may be excluded from training batches during the filtering process. However, we expect that the number of clean examples used for training would increase gradually as training proceeds since the proposed LEC allows the network to generalize clean examples without overfitting. To confirm this, we measured the recall defined by (# of clean examples used for training / # of all clean examples) during training. We observed that recalls of the proposed LEC sharply increase. The results can be found in Section 5.1 in our revised paper. In addition, pre-training [1] prior to the filtering process may help in reducing the number of clean examples removed by ensemble consensus filtering.
>
> Q4. The authors should clearly discuss the computational complexity and space complexity of their proposals.
>
> >> We added the complexity analysis to Appendix A.2. The computational complexity was computed by the number of forward and backward passes, and the space complexity was computed by the number of network parameters. Here, we assumed that early stopping was not used and the noise ratio was given. Note that the computational complexity of each method depends on its hyperparameter values, e.g., the noise ratio. In the following table, “M” indicates the number of networks used for filtering in LECs. Our proposed LTEC is the most efficient because it can be implemented with a single network based on Section A.1.3 and it updates only a subset of the entire set after the warming-up process.
>
> ====================================================================================
> 					 	      Standard   Self-training   Co-teaching    LNEC       LSEC      LTEC
> ====================================================================================
> space complexity
>   # of network parameters             m                  m                  2*m              M*m	         m             m
> ——————————————————————————————————————————————
> computational complexity
>           # of forward passes             n                    n                  2*n                  M*n         M*n          n
>        # of backward passes             n                  =<n             =<2*n               =< M*n       =< n       =< n
> ====================================================================================
>
> * Reference
> [1] Hendrycks et al., Using Pre-Training Can Improve Model Robustness and Uncertainty, ICML 2019.
>
> Thanks again for taking the time to review.

---

### Official Review · AnonReviewer4 · 2019-10-29
**Official Blind Review #4**

**Rating:** 8

**Review:**

Summary:
This paper proposes a general method for eliminating noisy labels in supervised learning based on the combination of two ideas: outputs of noisy examples are less robust under noise, and noisy labels are less likely to have a low loss. The authors then propose 3 concrete instantiations of the idea, and do a thorough empirical study (including ablations) across multiple architectures, datasets, noise types, and comparing to multiple related methods. The results show pretty convincingly that one of the new methods (LTEC) that uses past networks outputs to build an ensemble performs really well.

Caveats:
1) I’m an emergency reviewer and had less time to do an in-depth review.
2) While my research is sufficiently close to review the paper, I’m not an expert on label noise specifically, so I cannot comment much on novelty and related work questions.

Comments:
* The readability of the paper could be dramatically improved by reporting results visually (eg bar-plots) and moving all tables into the appendix.
* The authors state that peak performance is valid because it *could* have been found using a validation set -- then why not just do that, and report this early-stopping performance everywhere, instead of always two numbers (peak and final)?
* Please make the perturbation properties in section 3.2 more precise: do you mean “there exists a threshold and perturbation such that for some (x, y)”? Or “For any threshold and any perturbation then for all (x, y) it holds...”? Or something in-between?
* For the “competing methods” paragraph, please cite the relevant papers in the main text, not only in the appendix.
* “over 4 runs” did you randomize the data noise in each run (and in the same way for each method?), or only the network initialisation?
* Table 5 is cool, but it raises the question: is 1.1epsilon the best, or would performance keep going up?
* Looking at the actual implementation of LTEC (Algorithm 4), I cannot resist the thought that M=infinity could work even better (at no extra cost): just maintain a monotonically shrinking set of samples?

Minor comments:
- Define the threshold symbol in section 3.2
- Define M in Algorithm 1
- Define (initial) \mathcal{P}_0 in Algorithm 4
- Fig 2: can you clarify what green is, does it correspond to “self-training”?
- “label precision … does not decrease” -- well, not a lot, but it does decrease!
- Fig 1, Fig 2: set max y to 100
- Table 4: Include M=1 (which is self-training) for comparison

**Experience Assessment:**

I have read many papers in this area.

**Review Assessment: Checking Correctness Of Derivations And Theory:**

N/A

**Review Assessment: Checking Correctness Of Experiments:**

I carefully checked the experiments.

**Review Assessment: Thoroughness In Paper Reading:**

I read the paper at least twice and used my best judgement in assessing the paper.

---

> ### Author Response · Authors · 2019-11-15
> **Response to review#4**
>
> We are grateful for your detailed feedback. We would like to answer your questions as follows:
>
> Q1.  The readability of the paper could be dramatically improved by reporting results visually (eg bar-plots) and moving all tables into the appendix.
>
> >> We are thankful for the suggestions. Unfortunately, we did not have enough time to do what you suggested. Instead, we tried to put tables and figures, and their corresponding sections on the same page to improve the readability.
>
> Q2. The authors state that peak performance is valid because it *could* have been found using a validation set -- then why not just do that, and report this early-stopping performance everywhere, instead of always two numbers (peak and final)?
>
> >> Some studies [1-3] provide the accuracy at or near the last epoch only. To help comparison with the other methods, we reported the final as well as peak accuracy.
>
> Q3. Please make the perturbation properties in section 3.2 more precise: do you mean “there exists a threshold and perturbation such that for some (x, y)”? Or “For any threshold and any perturbation then for all (x, y) it holds...”? Or something in-between?
>
> >> Sorry for the confusion. We defined some notions in Section 3.1 and clarified Section 3.2 and 3.3 using the notions in our revised paper. Please read these sections.
>
> Q4. For the “competing methods” paragraph, please cite the relevant papers in the main text, not only in the appendix.
>
> >> Thank you for pointing it out. We cited the relevant papers in the main text as well as in the appendix.
>
> Q5. “over 4 runs” did you randomize the data noise in each run (and in the same way for each method?), or only the network initialisation?
>
> >> For each noise type, we ran every method four times with four seeds, e.g., four runs of Standard on CIFAR-10 with sym-20%. Random label noise and open-set noise were randomly generated. For each run of both random label noise and open-set noise, both a noisy dataset and network initialization were randomized. Here, four noisy datasets generated in four runs are the same for all methods. In our revised paper, results on *semantic noise* were added. Semantic noise was deterministically generated. For each run of semantic noise, only network initialization was randomized.
>
> Q6. Table 5 is cool, but it raises the question: is 1.1epsilon the best, or would performance keep going up?
>
> >> Table 5 shows that 1.1 epsilon is not the best on CIFAR-10 with sym-60%. We speculate that this is due to the reduction in the total number of samples used for training, which may be harmful to the performance.
>
> Q7. Looking at the actual implementation of LTEC (Algorithm 4), I cannot resist the thought that M=infinity could work even better (at no extra cost): just maintain a monotonically shrinking set of samples?
>
> >> The results (final/peak acc.) of LTEC with varying M on *CIFAR-10 with random label noise* are as follows:
>
> =================================================================================
>                              M = 1              M = 3               M = 5               M = 7               M = 9              M = ∞
> =================================================================================
> sym-20%   | 84.96/85.02 | 87.68/87.78 | 88.18/88.28 | 88.63/88.77 | 88.79/88.87 | 86.57/86.62
> sym-60%   | 73.99/74.35 | 79.73/79.80 | 80.38/80.52 | 80.39/80.45 | 80.28/80.39 | 71.63/71.86
> asym-20% | 85.02/85.24 | 87.85/88.15 | 88.86/88.93 | 88.96/89.07 | 88.99/89.11 | 85.55/85.59
> asym-40% | 78.84/79.66 | 85.44/85.59 | 86.36/86.50 | 86.78/86.82 | 86.59/86.63 | 77.30/77.40
> =================================================================================
>
> The results show that larger M does not always lead to better performance. In Figure A3, we observe that ensemble consensus filtering with too large M removes even clean examples from training batches *in the early stage of the filtering process*. We expect that pre-training [4] prior to “filtering process” helps in reducing the number of clean examples removed by ensemble consensus filtering regardless of “M”.
>
> Q8. Minor comments
> >> Thanks for pointing out some issues. We revised the paper based on the comments you provided.
>
> * Reference
> [1] Ma et al., Dimensionality-Driven Learning with Noisy Labels, ICML 2018.
> [2] Han et al., Co-teaching: Robust Training of Deep Neural Networks with Extremely Noisy Labels, NeurIPS 2018.
> [3] Yu et al., How does disagreement help generalization against label corruption?, ICML 2019.
> [4] Hendrycks et al., Using Pre-Training Can Improve Model Robustness and Uncertainty, ICML 2019.
>
> Thanks again for taking the time to review.

---

### Public Comment · ~Yanyao_Shen1 · 2019-10-29
**Relevant paper that explains why selecting small loss samples is reasonable**

Dear authors, thanks for uploading this interesting work.

In case you are not aware of, we have theoretically shown why selecting small loss samples is reasonable for robust training: http://proceedings.mlr.press/v97/shen19e/shen19e.pdf

We also provided a simple algorithm that works in many applications, including noisy label. Different from co-teaching, we do not need the dual network structure, but we re-train the network based on the selected 'good samples' from previous iteration (selecting step). Intuitively, since clean samples may generalize, by iteratively running the selecting step, more and more clean samples will be selected.

---

> ### Author Response · Authors · 2019-10-30
> **RE: Relevant paper that explains why selecting small loss samples is reasonable**
>
> Dear Yanyao Shen,
> Thank you for your interest and for introducing your work. We are happy to learn the theoretical investigation of training with small-loss examples from your work. In particular, your work is very similar to “self-training” used as the baseline in our study although their implementation details slightly differ (e.g., the number of selected samples, sampling from the entire set/mini-batch). We will cite your work in our revised paper. Thanks!

---

### Author Response · Authors · 2019-11-15
**Revision summary**

We have uploaded our revised paper. The major changes are as follows:

1. (Section 3.1, 3.2, and 3.3) We defined some notions in Section 3.1 and clarified Section 3.2 and 3.3 using the notions in our revised paper.

2. (Section 3.2 and 4.3, and Algorithm 2) We proposed “LEC-full” algorithm to encourage more stable training with *fixed* batchsize. We also reported their results on various noisy datasets.

3. (Section 4.1 and 4.3) We added results on CIFAR with *semantic noise* which is more realistic. The semantic noise is generated as follows:
We first select the top e% most uncertain images from CIFAR and then mislabel those selected images ambiguously.

4. (Section 5.1) We added the recall analysis to identify whether clean examples are excluded from training batches. Here, recall is defined by  (# of clean examples used for training / # of all clean examples).

5. (Section A.1.3) We modified pseudocodes for the proposed algorithms.

6. (Section A.2) We added the complexity analysis of the proposed algorithms.

We are thankful to reviewers for their help with the revision.

---

### Decision · Program_Chairs · 2019-12-19

**Decision:**

Accept (Poster)

**Comment:**

This paper proposes an ensemble method to identify noisy labels in the training data of supervised learning.  The underlying hypothesis is that examples with label noise require memorization.  The paper proposes methods to identify and remove bad training examples by retaining only the training data that maintains low losses after perturbations to the model parameters.  This idea is developed in several candidate ensemble algorithms.  One of the proposed ensemble methods exceeds the performance of state-of-the-art methods on MNIST, CIFAR-10 and CIFAR-100.

The reviewers found several strengths and a few weaknesses in the paper.  The paper was well motivated and clear.  The proposed solution was novel and plausible.  The experiments were comprehensive.  The reviewers identified several parts of the paper that could be more clear or where more detail could be provided, including a complexity analysis and
extended experiments.  The author response addressed the reviewer questions directly and also in a revised document.  In the discussion phase, the reviewers were largely satisfied that their concerns were addressed.

This paper should be accepted for publication as the paper presents a clear problem and solution method along with convincing evidence of method's merits.